# The Consciousness of Pain: A Thalamocortical Perspective

**Paraskevi Sgourdou**

Department of Genetics, Perelman School of Medicine, University of Pennsylvania, 3700 Hamilton Walk, Philadelphia, PA 19104, USA; paraskevi.sgourdou@pennmedicine.upenn.edu or psgourdou@gmail.com

**Abstract:** Deep, dreamless sleep is considered the only "normal" state under which consciousness is lost. The main reason for the voluntary, external induction of an unconscious state, via general anesthesia, is to silence the brain circuitry of nociception. In this article, I describe the perception of pain as a neural and behavioral correlate of consciousness. I briefly mention the brain areas and parameters that are connected to the presence of consciousness, mainly by virtue of their absence under deep anesthesia, and parallel those to brain areas responsible for the perception of pain. Activity in certain parts of the cortex and thalamus, and the interaction between them, will be the main focus of discussion as they represent a common ground that connects our general conscious state and our ability to sense the environment around us, including the painful stimuli. A plethora of correlative and causal evidence has been described thus far to explain the brain's involvement in consciousness and nociception. Despite the great advancement in our current knowledge, the manifestation and true nature of the perception of pain, or any conscious experience, are far from being fully understood.

**Keywords:** consciousness; pain; anesthesia; thalamus; cortex





## 1. Introduction

For a long time, the concept of consciousness was one from which science remained distant, partially due to its inability to give an accurate definition of consciousness, as well as the absence of effective ways to study it. However, recent technological and scientific advancements, including the rapid development of artificial intelligence (AI), have allowed, or even mandated, the scientific pursuit of the nature of consciousness. More importantly, the multidimensionality of human existence and the uniqueness of every conscious experience have been the subject of contemplation for many philosophers and scientists throughout history. Since the time of René Descartes, and the introduction of the classic dualism, there has been a clear distinction between mind and body. Their definition as distinct substances, albeit their postulated interaction, allowed science to pursue the study of the physical aspect of existence (body) and leave the non-physical (mind) to religion and philosophy. Different forms of dualism, whether they propose a causal interaction between mind and body (Descartes's dualism) or consider them preexisting, separate substances (Gottfried Leibniz's or Thomas Huxley's epiphenomenalism), persist in modern philosophy. For example, the modern-day philosopher Saul Kripke has argued that although various mental phenomena, like pain, have neural correlates they are not identical to them.

According to a basic, dictionary description, consciousness can be defined as "an alert cognitive state in which one is aware of themselves and the environment around them". Within this simple definition, consciousness includes the dimensions of wakefulness (level of consciousness) and awareness (context of consciousness) (Figure 1). Both need to coexist in order for conscious experience to manifest. Exceptions include the state of lucid dreaming, during which a level of awareness exists in the absence of wakefulness [1], as well as several disorders of consciousness (DOC), in which various defects of awareness exist in the presence of wakefulness [2,3]. Furthermore, the concept of consciousness

cannot be reduced to the conscious perceptions, as subjective experiences, but also needs to include the ability to remember them as well as a general sense of selfhood [4]. One equally important aspect of consciousness is the ability to be conscious of our consciousness, and its nature (meta-consciousness) [5]. Many additional properties have been attributed to the general concept of consciousness (e.g., intentionality, pleasantness, familiarity, Gestaltness, etc.), not all of which co-exist in every conscious experience [6].

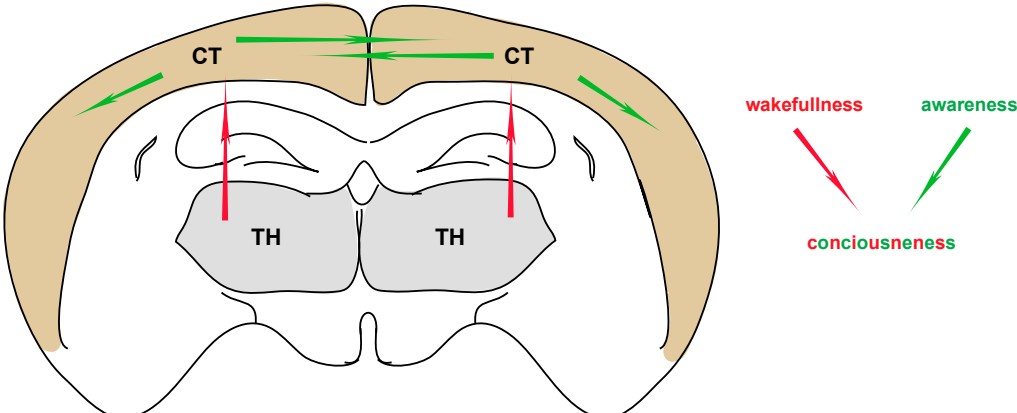

**Figure 1. The basic parameters of consciousness.** Corticocortical interactions are generally thought to contribute to the aspect of awareness, whereas thalamocortical interactions contribute to the aspect of wakefulness. Both are necessary for manifesting a "normal" state of consciousness. TH: thalamus, CT: cortex.

Similar to consciousness, the origin and nature of the perception of pain has historically been the subject of multiple theories [7]. Plato and Aristotle defined pain as an emotion, a "passion of the soul" [8]. The father of Western medicine on the other hand, Hippocrates, who was the first to separate medicine from religion, believed that all diseases occurred naturally. He thought they were the result of environmental factors, diet and living habits, instead of superstition or punishment inflicted by gods, and postulated that pain was a result of imbalance in the humors [9]. Later on, Galen, the physician of the second century AD whose work dominated Western medicine for many centuries, thought that pain was always the result of physical injury [10]. During the scientific revolution of the 17th century, it was René Descartes who explained the pathophysiology of pain in a mechanistic way, describing an ascending pathway in which the stimulus from the periphery was transmitted to the spinal cord and then to the pineal gland where the conscious experience was formed [11]. Descartes's deductive theory tried to connect the physiological aspect of nociception with the conscious percept of pain.

Today's definition of pain, as given by the International Association for the Study of Pain (IASP), is "an unpleasant sensory and emotional experience associated with actual or potential tissue damage, or described in terms of such damage" [12]. Although pain is mostly caused by noxious stimuli, there are cases when innocuous stimuli can cause pain (allodynia, hyperalgesia), as well as cases of chronic pain. Like consciousness, pain is a percept of multiple elements that make it more complex than nociception, which only refers to the "neural process of encoding noxious stimuli". Except for the physical, various other psychosocial factors can influence the perception of pain and the transition from acute to chronic pain. Additionally, the latter can be felt even in the complete absence of any tissue damage or physical stimulus, like in neuropathic pain [13,14]. Therefore, the intensity of a noxious stimulus is not necessarily directly analogous to the experience of pain, and in some instances, nociception can exist without conscious pain awareness. For example, nociceptive signals have been detected in patients under anesthesia, a further indication that any form of activity measurement can only serve as an inference of pain [15]. The fact that there is tremendous variability in the way individuals experience pain poses a lot of challenges for both the research and clinical applications. Even more complicated, and clinically challenging, becomes the assessment and

treatment of pain in patients with disorders of consciousness (DOC) due to several ethical considerations and controversies involved [16]. Furthermore, the observations made about neuronal activity in certain parts of the brain during nociception do not automatically prove that engagement of these brain regions is necessary for the perception of pain, adding another layer of complexity [17–19]. Here, I mainly focus on pain as part of the general consciousness, which is lost under anesthesia. I briefly describe the thalamocortical perspective of both pain and consciousness, as a common neural correlate between them.

## 2. Inducible Unconsciousness

### 2.1. Loss of Consciousness and Anesthesia

Although there is a great debate on whether anesthesia and the effects it has on brain activity can really lead to loss of consciousness, there are two main hypotheses proposed thus far to explain how anesthesia interferes with consciousness. The first hypothesis supports the disruption of feedback connectivity amongst corticocortical loops [20], most likely through the downregulation of the distal dendrites of Layer 5 pyramidal neurons that extend upwards into Layer 1, where they receive cortical feedback connections [21]. The second hypothesis points more to the disruption of higher order thalamocortical loops [22]. There is, however, room for integrating the "bottom-up" mechanism (thought to be responsible for repressing the level of consciousness) and the "top-down" mechanism (thought to deregulate the content of consciousness) via which anesthesia is working, in a model where the two pathways are not separate but interactive [23]. Lawrence M. Ward [24] has even proposed the *thalamic dynamic core* theory, according to which "phenomenal consciousness is generated by synchronized neural activity in the dendritic trees of dorsal thalamic neurons". Such activity is greatly reduced in studies of various general anesthetics. Regardless of the mechanism, the brain's ability to effectively integrate information is blocked.

The effect of anesthesia on consciousness seems to be gradual, as it is increasing with the dose of anesthetics. Initially, it suppresses thinking, attention and memory, followed by a gradual loss of voluntary responsiveness and finally suppression of nociceptive and autonomic reflexes [25]. General anesthesia is the only voluntarily induced state of unconsciousness during which the circuits of pain are suppressed (analgesia). Except for inducing analgesia, anesthesia is also suppressing movement (akinesia) and memory (amnesia) during the time it is applied, all of which are reversible effects. There are, however, instances when general anesthesia can produce states of disturbed consciousness, like delirium or several behavioral excitation states ranging from hyperlocomotion to schizophrenia-like symptoms [26,27]. The mechanisms of these brain states are largely unknown and will not be addressed here.

### 2.2. Thalamocortical Connections in Anesthesia

The induction of anesthesia has traditionally been linked with inhibition, given that the vast majority of anesthetics are known to upregulate GABA$_A$ receptors [28]. Several studies have shown a "top-down" mechanism of inducing unconsciousness via a direct effect on cortical neurons. These effects include the suppression of excitatory action potentials [29,30] and the inhibition of GABAergic interneurons, thereby disinhibiting pyramidal cortical neurons [31,32]. However, both cortical and subcortical areas were shown to be important for modulating consciousness, the latter by decreasing arousal and the former by compromising the contents of consciousness [23]. Except for several hypothalamic nuclei known to play a role in anesthesia and sleep, the reticular thalamic nucleus (TRN) has also been shown to suppress cortical activity during isoflurane anesthesia [33] and increase sensitivity to propofol via GABA$_B$ receptors [34]. Epithalamic nuclei tightly connected to the thalamus, like the lateral habenula LHb, are activated by several anesthetics as well [35]. Interestingly, in many studies of arousal and emergence from anesthesia, activation of the central medial thalamus (CM) is facilitating the process, not surprisingly so, given that anesthetics inhibit neurons within CM [36].

Compromised functional thalamocortical connectivity is known to induce a state of unconsciousness [37]. While a substantial amount of activity is maintained between cortical regions during both sleep and light anesthesia, thalamocortical communication is lost. A distinction has been made between the intralaminar thalamus, which includes the non-specific midline nuclei, and the specific nuclei in other parts of the thalamus. More specifically, the former has been linked to consciousness via the integration of cortical information, while the latter is responsible for the transmission of motor and sensory signals [38]. Studies in humans [39] and rats [40] have shown that the intralaminal thalamus is the one affected first by anesthesia followed by changes in the specific nuclei. In non-human primates, the central lateral thalamus (CL) in particular, and its influence in deep layer activity, was shown to play a more important role in consciousness compared to neighboring midline thalamic nuclei, like the mediodorsal (MD) and centromedial (CM) nucleus [41].

Similarly, propofol-induced anesthesia causes profound changes in cortical physiology that are attributed to the loss of functional thalamocortical connectivity [42]. More specifically, the ventral intermediate nucleus (ViM) has been shown to share structural and functional connectivity with the ipsilateral sensorimotor cortex, both of which undergo changes in local power during induction of propofol anesthesia. The authors suggest an intrathalamic pathway between ViM (a motor related nucleus) and sensory-specific thalamic nuclei, rather than the interaction between motor and sensory cortices, to explain the observed functional connectivity. This explanation could also apply to the connection between the intralaminal and the specific thalamic nuclei, both of which are affected during anesthesia. Recently, a study by Suzuki M. and Larkum M.E. [43] provided evidence for a theory that reconciles the two distinct hypotheses about the way anesthesia affects consciousness. The authors showed that in anesthetized mice the coupling between the distal dendrites and the somata of Layer 5 somatosensory neurons is impaired. Furthermore, this coupling was greatly dependent on input from the higher order posterior thalamic nucleus (POm). The integration of the two main theories regarding anesthesia can also explain the seemingly paradoxical mode of action of many anesthetics, which have little effect on neuronal firing per se, but might influence the integration of information (or phi, see [44]) in the brain [45].

### 3. The Percept of Pain

#### 3.1. Pain and Consciousness

Pain, similar to other senses, is an experience of unique multidimensionality. Except for the basic somatosensory, discriminatory aspect of pain, there are also the affective and cognitive aspects of the experience, all of which require a certain level of consciousness (wakefulness and awareness). In general, it is thought that the perception of pain increases with the level of consciousness. The relationship between pain perception and our conscious state becomes more evident in studies like the one by Jensen et al. [46], which showed that subliminal cues can act in ways to either enhance or reduce pain responses depending on the context with which those cues were previously associated (painful high temperature stimulus versus innocuous low temperature stimulus). Similarly, mental imagery of pain-related or "coping" images can exacerbate or alleviate pain, respectively, and even reduce chronic pain or phantom pain experienced by amputees [47]. The fact that non-pharmacological treatments and/or complementary and alternative approaches have the ability to influence the physical perception of pain, reinforces the direct connection of pain and consciousness.

As there is no "brain center" for consciousness, in a similar way, there is no "brain center" for pain. Three main, temporally distinct phases have been described to constitute conscious nociception [48]. The first one involves the pre-conscious brain activities (reported in the posterior insula, operculum, mid-cingulate and amygdala), the second is defined by conscious voluntary reactions (with activation of the antero-insular, prefrontal and posterior parietal cortices), and the last one (involving persistent activity in the

hippocampus, perigenual and perisplenial cingulate cortices) is thought to play a role in high-level stages of nociceptive processing that link the "external" and "internal" sensory worlds. It is important to note that the thalamus and somatosensory cortex were not examined in this study. Interestingly, various practices of mindfulness, like meditation, have been shown to integrate these "intrinsic" and "extrinsic" aspects of pain, a skill that can be enhanced through greater practice [49]. Some evidence shows that such practices probably act through deactivating the precuneus/posterior cingulate cortex (PCC) which is known to facilitate integration of extrinsic, sensory with intrinsic, self-narrative processes [50,51]. The networks responsible for these two processes are functionally disconnected in patients with altered states of consciousness like the vegetative state [52].

While nociceptive processing is mostly unconscious, several psychological, socio-cultural and cognitive (e.g., attention) factors interact in order to produce the percept of pain [49]. An interesting effect of cognition on pain perception is shown by the heightened pain response to novel nociceptive stimuli, known as the pain alarm response or "surprise effect", which links conscious awareness with the response to pain [53]. When trying to theoretically distinguish, and even practically separate, the informational (sensory) from the affective aspect of pain, the example of cingulotomy is truly fascinating. Cingulotomy describes the surgical removal of the anterior cingulate cortex in patients with chronic, excruciating pain, resulting in them retaining the sensory but losing the affective component of pain, an effect quite similar to morphine administration. In other words, they feel the pain but do not mind it (asymbolia) [54]. Clearly, by altering the brain's connectivity we can "reduce" the experience of pain, and possibly other conscious experiences as well, to a basic phenomenal awareness.

### 3.2. Thalamocortical Connectivity in Pain

Several fMRI studies in humans have shown consistent activation of the ventrolateral thalamus, secondary somatosensory cortex, midcingulate cortex and the dorsal posterior insula during physical pain [55], regardless of stimulation technique or location of induction [56]. Sensory inputs coming to the thalamus via the spinothalamic track during nociception terminate in the somatosensory ventral posterior lateral thalamic nucleus (VPL), the ventral posterior inferior nucleus (VPI) and the posterior group of thalamic nuclei [57]. An area within the thalamus called the posterior extension of the ventral medial thalamic nucleus (VMpo), which is not a typical nucleus, connects the lateral to the posterior thalamic areas that are innervated by the spinothalamic track. VMpo has been shown to play an important role in nociception and thermoception [58,59]. The limbic system, on the other hand, has been well established for playing a key role in the emotional, affective aspect of pain as well as the transition to chronic pain. Except for the limbic system, central thalamic nuclei are also involved in the emotional and aversive quality of pain. The mediodorsal thalamus (MD), central lateral nucleus (CL) and parafasicular nucleus (PF), all project to both anterior and motor areas of the cingulate cortex as well as the PFC [60]. As a general model, the lateral thalamocortical sensory system is considered responsible for the discriminative aspect of pain, the medial for the motor and suffering (cognitive, emotional and autonomic) aspects of pain, and the posterior for the initial percept of nociception and its intensity.

The thalamocortical circuitry in rodents has similar characteristics. Somatosensory information is transferred to the primary somatosensory cortex (S1) via the VPM/L thalamic nuclei. In turn, Layer 5b and Layer 6 neurons within S1 project to the secondary, higher order POm, the reticular thalamic nucleus and areas outside the thalamus [61–63]. POm sends projections back to S1, as well as to the secondary somatosensory cortex (S2) and primary motor cortex (M1), thereby controlling the motor responses to pain. Measurements with eight-wire microarrays in four different brain areas (S1, anterior cingulate cortex ACC, ventral posterior thalamus VP and medio-dorsal thalamus MD) in rats, revealed an increased information flow from S1 to VP upon noxious heat stimulation of the hindpaw [64]. The information flow was taking place both in the ascending (thalamus to cortex) and

descending (cortex to thalamus) directions, with the medial pathway (MD-ACC) harboring more diffuse, multisynaptic connections in comparison to the tight and fast monosynaptic connections of the lateral (VP-S1) pathway. This might indicate the slower and prolonged emotional/aversive response to pain, which is partially mediated by the medial thalamic nuclei [65].

## 4. Conclusions and Perspective

Consciousness, as a basic prerequisite for the perception of pain, shares with it several brain neural correlates. Here, I have briefly described the common correlates involving activity within, and interactions between, the cortex and thalamus. Although we are far from fully understanding the exact mechanisms of pain and, even less so, the emergence of consciousness, it is clear that the cortex and thalamus are structures that play key roles. Both consciousness and pain are lost under general anesthesia. Even though the application of general anesthesia has the only purpose of deactivating the neuronal circuits of pain, there are many other aspects of consciousness that are blocked during anesthesia. The effort to unravel the mechanisms of action of the main anesthetics used today, will help scientists come closer to an explanation of consciousness. At the same time, the study of pain can be very informative about the unique and enigmatic nature of consciousness, whose different states exert significant influence on the subjective pain experience. The different phases of pain have been described in full vigilance states, in relation to the progression from nociception to the conscious perception of pain [48]. Interestingly, even in unconscious states, the vegetative activation by nociceptive stimuli via spino-amygdala pathways can produce some level of implicit memory, which possibly contributes to the development of hyperalgesia syndromes in coma survivors [66]. Therefore, it becomes apparent that the way the brain turns nociceptive signals into conscious pain perception is far more complex.

### 4.1. Pain: Evolution, Adaptation and Control

As a conscious experience, pain is clearly one that we would like to avoid or at least alleviate to a great extent. This is solely a result of the limbic activation caused by noxious stimuli, which creates an aversion to anything potentially unpleasant. Would the absence of pain, however, be beneficial to the long-term survival and evolution of our species? The clear answer is certainly not. The most direct indication of that are the people suffering from congenital syndromes of pain deficiency who have lost the capacity to experience pain, resulting in excessive tissue damage and highly reduced life expectancy [67]. In cases of drug-induced analgesia, the effect is also oftentimes negative, as shown for example by the faster damage and joint deterioration in patients with osteoarthritis using anti-inflammatory drugs [68]. Evidently, pain is necessary for preventing further tissue damage, promoting healing and alerting one to potential danger. While acute pain is an adaptation mechanism for a species' survival, chronic pain becomes highly maladaptive with a host of negative effects on the psychology and physiology of an organism [69]. Interestingly, hypersensitivity to pain (allodynia) is far more prevalent than the syndromes of pain deficiency. This makes evolutionary sense, given that the former, albeit the undoubtable discomfort and agony it causes, is much less detrimental to the organism's survival than the latter.

Apart from the genetic syndromes of pain deficiency, humans have the capacity to "silence" the perception of pain by various forms of mindfulness, leading to placebo analgesia [70]. This control is exerted by activation of the descending pain inhibitory pathway that balances the two ascending pain pathways, lateral and medial, via which we are experiencing the discriminatory/sensory and affective/motivational aspects of pain, respectively. The people who can walk on charcoals without being burned, or the martial artists who break concrete surfaces without feeling physical pain, are characteristic examples of such acute pain control. The puzzling observation of "mind over matter" has been a phenomenon for which science has limited mechanistic explanation. In addition to beliefs and expectations influencing the conscious experience of pain, they can also shape

the perception of unpleasantness associated with a nociceptive sensory input. Furthermore, the contextual evaluation of pain can influence to a great extent the affective dimension of the experience. For example, the way a cancer patient is experiencing pain, even if the sensory aspect of it is of average intensity, is fundamentally different to the labor pains experienced by a mother. In the latter case, the affective aspect of pain becomes more bearable while the intense pain is being "overridden" by the anticipation of a new life, as opposed to a patient who is having to cope with the pain of a condition that is potentially threatening their life [71]. The comorbidity of chronic pain and anxiety is a common phenomenon and the shared underling brain circuitry a biological indication of their close interaction [72]. Contextualization, however, can work both in negative and positive ways, based not only on the factual part of reality but also on the way we choose to perceive it.

*4.2. Pain as a Conscious Experience*

If a conscious being, like ourselves, can really shape reality, one can only wonder to what extent the "observer" can influence any conscious experience including pain. Consciousness, as one of the most important "problems" of science, and human existence in general, is a topic where neuroscience and quantum physics meet in an effort to explain reality. Taking into consideration the famous phrase by Stephen Hawking stating that "the past, like the future, is indefinite and exists only as a spectrum of possibilities", it would be possible for us to imagine that a conscious entity with the capacity for memory would be able to "choose" their conscious experiences. In a way, this is how the placebo and nocebo effects come to be, and possibly how all the "mind over matter" phenomena actualize. Science has still little understanding of these effects, but their existence is acknowledged and documented, particularly in relation to modifying the conscious experience of chronic pain [49]. Interestingly, this view seems to be in accordance with panpsychism, a theory which postulates that consciousness exists in everything and is the fundamental feature, in essence the generator and shaper, of reality. One of the most prevalent theories of consciousness today, the information integration theory (ITT), proposed by Giulio Tononi, is considered (or criticized) by some thinkers to "imply" panpsychism as well. ITT tries to explain mathematically the parameters that dictate the extent to which any given system is conscious (quantity) and the kind of consciousness it possesses (quality) [44]. According to the author who proposes that "consciousness is a fundamental quantity, that it is graded", any system can possess a level of consciousness that is depended on its ability to integrate information (the phi factor). The higher this ability, the higher the consciousness. One empirical, biological observation that supports this theory is the example that damage in the cerebellum does not significantly alter a patient's conscious experience, as opposed to damage in other parts of the brain (e.g., cortex, thalamus) [73,74]. This is attributed to the fact that the cerebellum has much less internal neuronal connectivity compared to other brain areas with increased interconnectivity, and therefore capacity to integrate information. In theory, any "entity" with basic interactive subunits (possibly even a photon with its interacting quarks) can have a certain number of phi property, and therefore a certain level of consciousness, hence the correlation of ITT with panpsychism.

Regardless of the theory, it remains clear that the concept of consciousness is not reducible to the mental/brain states (objective), since it has a first-person ontology (subjective). The scientific endeavor regarding the nature of conscious experiences, including pain, is focusing on finding correlations which will hopefully lead to causal relations. The multidimensionality and subjectivity of pain make it a unique sense modality, the study of which can give us insights into the biological substrate of the conscious experience in general. Many scientists and/or philosophers believe that it is just a matter of time, and thereby of advancement in technology, until we can fully demystify the concept of consciousness. Others claim that we will never be able to completely dissect, or accurately define, consciousness even if we manage to acquire knowledge about all its neural correlates and the interactions amongst them. We might be able to describe all the physical processes in great detail, but we would still not manage to explain the nature of a subjective

experience like that of pain. In other words, we will not solve the so-called (by philosopher David Chalmers) "hard problem" of consciousness, even if/when we completely solve the "easy problem". Either way, the joy of the quest itself is worth the while.

After all, the price of being conscious is feeling the pain.

**Funding:** This research received no external funding.

**Institutional Review Board Statement:** Not applicable.

**Informed Consent Statement:** Not applicable.

**Data Availability Statement:** Not applicable.

**Acknowledgments:** In loving memory of my father Panagiotis A. Sgourdos.

**Conflicts of Interest:** The author declares no conflict of interest.

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
