# Peer review of "The Consciousness of Pain: A Thalamocortical Perspective"

_neurosci, doi:10.3390/neurosci3020022_

Round 1

Reviewer 1 Report

The manuscript presented by Paraskevi Sgourdou, entitled "The consciousness of pain: a thalamocortical perspective" is interesting and original. The article is well written, however I would suggest the author to insert a table with all previous studies mentioned in the text and presented in literature (to improve the interest of reading).      

Author Response

Thank you for your time reviewing the manuscript, and for your comments.

I agree that tables are in general helpful, however, in this instance several studies are mentioned in the paper (in essence all references) that are pertinent to different aspects of the subjects (consciousness and pain) addressed. Therefore, in this manuscript, it would probably be impractical.

Reviewer 2 Report

Review for NeuroSci (May 13th, 2022)

Title: The Consciousness of pain: a thalamocortical perspective

Summary:

In their article entitled “The Consciousness of pain: a thalamocortical perspective”, the author delineates the neuroanatomy of pain perception, its relevance to consciousness, and the use of anesthetics as a means of parsing new information about these topics. The review is substantive with plenty of context and will be useful to investigators in the field. Unique insights are provided that add significantly to the review. There are many minor spelling and grammatical errors; I have identified many but I’m sure there are more so the paper should be looked over by another reader. Most of my concerns are minor; however, some major concerns are detailed below. Overall, this is an interesting paper that should be considered for publication.

Major comments:

- Line 76-77: Here, the author suggests that consciousness is a percept of multiple elements, which differs somewhat from their original definition involving wakefulness and awareness; is this intentional? Clarification may be required.

- Line 118-119: Would the author consider the consumption of opiates and other non-anesthetic compounds as “voluntarily induced states of unconsciousness” during which “the circuits of nociception are suppressed”? Also, I wonder whether the statement should be changed to “the circuits of pain are suppressed” given peripheral nociception may be ongoing despite the use of a general anesthetic (i.e., capsacin may activate an ion channel peripherally and trigger a nociceptive pathway but not be experienced while under the influence of a general anesthetic).

- It would be useful to know if there are any contributions from glial cells associated with the consciousness of pain and its interaction with anesthetics. This seems relevant due to cortical involvement.

- Line 134-153: The variability of neuroanatomical involvement is fascinating. Do we know why different anesthetics affect different brain regions? Is it a matter of different receptor types or densities? Myelin distribution? Vascular penetrance? Cell types? Some mechanistic rationale would be useful.

- Line 177-178: What does the author mean by “level of consciousness”. Level of wakefulness? Awareness? What is level referring to and what are its dimensions?

- Has the author considered discussing the well-known exacerbation of pain perception associated with anxiety (over-rating pain when anxiety is high)? The relevant neuroanatomy (e.g., orbitofrontal regions, amygdala, prefrontal cortices, limbic regions) would likely add significantly to the paper.

- I wonder how a neuroanatomical approach to the description of the phenomenon of pain would ever achieve something like a complete understanding since it does not address the subjective dimension, which is the thing we are really trying to understand. Perhaps the author should spend some time describing the limitations of the neuroanatomical approach for this and other reasons. I see the final paragraph contains some description of this (near the very end) but reference to limitations of certain approaches may be useful.

- Supporting statements on Line 299-301, the author can cite ritual amputations and rites of passage involving pain (e.g. ant bites, spear competition, lashing) for which some cultures claim no pain perception; this will need to be referenced if included

Minor comments:

- In the title, “a” following the colon should be capitalized (at editor’s discretion)
- Line 26 “the scientific pursue” should be “the scientific pursuit”
- Line 27-28 “consciouss”  should be “conscious”
- Line 30-31 “Albeit their postulated interaction, them being defined as distinct substances allowed science”, the beginning of this sentence does not make much sense; there may be a missing comma or word?
- Line 58 “Similarly to consciousness” should be “Similar to consciousness”
- Line 59 “Arostitle” should be “Aristotle”
- Line 62-64 The sentence that begins with “He thought…” has too many instances of “and”; consider breaking it up or adding punctuation if possible
- Line 70 “Descarte’s” should be “Descartes’” (i.e., with the apostrophe after the “s”)
- Line 70 “..deductive theory, tried”; the comma can be removed here – it is not necessary
- Line 81 “like in neuropathic pain” – I agree with this sentence/statement but it should be cited
- Line 120: “akynisia” should be “akinesia”
- Line 122: the sentence beginning with “e.g. …” should be in parentheses or the sentence can be revised
- Line 128, please provide a citation/reference for the claim about upregulation of GABA-A receptors; interesting if true!

- Line 130: Action potentials are not “excitatory” or “inhibitory”; they can either be associated with downstream excitation or inhibition (depending on their postsynaptic effects on membrane potential) but action potentials are always “excitatory” by definition (they are evidence of a membrane that has become depolarized or “excited”); this is a minor issue but should probably be clarified; perhaps the inhibition of excitatory neurons? Or perhaps the suppression of action potentials?

- Line 168 “depended” should be “dependent”
- Line 169: “POm" is undefined?
- Line 169 “regarding anesthesia, can”; the comma can be removed
- Line 171: mention of phi may be confusing for readers unfamiliar with IIT; it may be useful to point the reader in the direction of the relevant literature here (i.e., “see X”); I realize that reference 41 does this implicitly but there should be some explicit mention.
- Line 175: the sentence that begins with “Pain, similarly to other senses” is maybe poetic to a fault; of course all experiences are multidimensional, including pain. I’m not sure the sentence adds much; consider changing but this is a minor issue

- Line 176 “discremenatory” should be “discriminatory”

- Line 241: POm is being defined here but was already mentioned on Line 169; either remove IT or define it on Line 169 and continue with the abbreviation
- Line 246 “reveiled” should be “revealed”

- Line 287 “prevalant” should “prevalent”

- Line 288 “undoughtable” should be “undoubtable”

- Line 289 “disscomfort” should be “discomfort”

- Line 333 – 335: Provide a reference for the claim about cerebellum damage

- Line 342 “reducable” should be “reducible”

- Line 344 “endenvor” should be “endeavor”
